# Risk factors for obstructed labour in Eastern Uganda: A case control study

Milton W. Musaba[1,2,3☯]*, Grace Ndeezi[3☯], Justus K. Barageine[4,5☯], Andrew Weeks[6☯], Victoria Nankabirwa[7,8‡], Felix Wamono[9‡], Daniel Semakula[5‡], James K. Tumwine[3‡], Julius N. Wandabwa[2☯]

1 Department of Obstetrics and Gynaecology, Mbale Regional Referral and Teaching Hospital, Mbale, Uganda, 2 Department of Obstetrics and Gynaecology, Busitema University Faculty of Health Sciences, Mbale, Uganda, 3 Department of Paediatrics and Child Health, School of Medicine, Makerere University College of Health Sciences, Kampala, Uganda, 4 Department of Obstetrics and Gynaecology, School of Medicine, Makerere University College of Health Sciences, Kampala, Uganda, 5 Africa Centre for Systematic Reviews and Knowledge Translation, Makerere University College of Health Sciences, Kampala, Uganda, 6 Sanyu Research Unit, University of Liverpool, University of Liverpool/Liverpool Women's Hospital, Liverpool, England, United Kingdom, 7 Department of Epidemiology and Biostatistics, School of Public Health, Makerere University, Kampala, Uganda, 8 Centre for Intervention Science and Maternal Child health (CISMAC), Centre for International Health, University of Bergen, Bergen, Norway, 9 School of Statistics and Planning, College of Business and Management Sciences, Makerere University, Kampala, Uganda

☯ These authors contributed equally to this work.
‡ These authors also contributed equally to this work.
* miltonmusaba@gmail.com

**Data Availability Statement:** All relevant data are within the manuscript and its Supporting Information files.

## Abstract

### Introduction

Obstructed labour (OL) is an important clinical and public health problem because of the associated maternal and perinatal morbidity and mortality. Risk factors for OL and its associated obstetric squeal are usually context specific. No epidemiological study has documented the risk factors for OL in Eastern Uganda. This study was conducted to identify the risk factors for OL in Mbale Hospital.

### Objective

To identify the risk factors for OL in Mbale Regional Referral and Teaching Hospital, Eastern Uganda.

### Methods

We conducted a case control study with 270 cases of women with OL and 270 controls of women without OL. We consecutively enrolled eligible cases between July 2018 and February 2019. For each case, we randomly selected one eligible control admitted in the same 24-hour period. Data was collected using face-to-face interviews and a review of patient notes. Logistic regression was used to identify the risk factors for OL.

### Results

The risk factors for OL were, being a referral from a lower health facility (AOR 6.80, 95% CI: 4.20–11.00), prime parity (AOR 2.15 95% CI: 1.26–3.66) and use of herbal medicines in

**Funding:** Survival Pluss Project at Makerere University. Funded by NORHED under NORAD. UGA-13-0030, Prof. James K. Tumwine. The funders had no role in study design, data collection and analysis, decision to publish, or preparation of the manuscript.

**Competing interests:** The authors have declared that no competing interests exist.

active labour (AOR 2.72 95% CI: 1.49–4.96). Married participants (AOR 0.59 95% CI: 0.35–0.97) with a delivery plan (AOR 0.56 95% CI: 0.35–0.90) and educated partners (AOR 0.57 95% CI: 0.33–0.98) were less likely to have OL. In the adjusted analysis, there was no association between four or more ANC visits and OL, adjusted odds ratio [(AOR) 0.96 95% CI: 0.57–1.63)].

## Conclusions

Prime parity, use of herbal medicines in labour and being a referral from a lower health facility were identified as risk factors. Being married with a delivery plan and an educated partner were protective of OL. Increased frequency of ANC attendance was not protective against obstructed labour.

## Introduction

Obstructed labour (OL) occurs when the foetal presenting part fails to descend despite adequate uterine contractions[1]. The global prevalence varies from 2–8%, being highest in low resource settings and almost none existent in high resource settings[1,2]. In Uganda, 8% of all maternal deaths (MDs) and 90% of perinatal deaths due to birth asphyxia are directly attributed to OL[3]. Almost three quarters of the MDs due to primary postpartum haemorrhage(PPH) and sepsis have OL as an underlying cause[4,5]. Limited or no access to quality emergency obstetric care services in low resource settings contributes to the high number of adverse obstetric outcomes[6].

Prevention of OL requires a multidisciplinary approach aimed in the short term at identifying high risk cases. In the long term,improving incomes at the level of the household would promote access to better nutrition, education and healthcare for the girl child [1,7]. Current evidence shows that access to skilled care during pregnancy and childbirth can mitigate adverse maternal and perinatal outcomes associated with OL[8]. In this regard, risk profiling during antenatal care (ANC) and intrapartum maternal fetal surveillance using a partogram are key interventions for early detection and management.

In Uganda, the utilisation of maternity services has improved with more than 90% for the first ANC visit, 60% for at least four ANC visits and facility births are at 73%[9]. Unfortunately, these improvements have not translated into a significant reduction in morbidity and mortality[9]. In addition, the known risk factors for OL have a poor predictive value that makes primary prevention difficult[10–12]. Parity, place of residence and age were significantly associated with OL after a review of patient records in six health facilities of western Uganda [2]. In Mbale Hospital, anecdotal evidence suggests that OL is the most common indication for primary emergency caesarean section and a cause of significant morbidity and mortality. The risk factors for OL and its associated obstetric sequel are usually context specific[13]. Currently, no epidemiological study has documented the risk factors for OL in Eastern Uganda. This study identified the risk factors for OL in Mbale Regional Referral and Teaching Hospital, Eastern Uganda. We hypothesised that increased frequency of ANC attendance (<4 versus $\geq$ 4 visits) was protective of OL.

## Materials and methods

### Study setting

We conducted this study in the labour suite of Mbale regional referral Hospital in Eastern Uganda. This hospital, serves 14 districts in the Elgon zone with an estimated population of 4

million people. This is a government run, not-for-profit, charge-free, 470-bed hospital with 52 maternity beds. Annually, about 12,000 childbirths occur in this hospital with a caesarean section rate of 35% and nearly 500 mothers have OL. About two thirds of these mothers with OL are referrals in active labour from the lower health units.

### Study design

Unmatched case control design with incidence density sampling of the controls admitted in the same delivery suite.

### Study population

All patients admitted to the labour suite in active labour at term ($\geq$ 37 weeks of gestation) were screened. A Medical Officer or Obstetrician diagnosed OL using the American College of Obstetricians and Gynecologists (ACOG) guideline for arrest of labour [14] and local protocols. A case was defined as; a cervical dilatation $\geq$ 6cm with ruptured membranes, having adequate contractions lasting > 4hrs with no change in cervical dilatation in the first stage of labour. For the second active stage of labour, arrest was defined as a delay of > 2 hours for the nullipara and > 1 hour for the multipara with adequate uterine contractions. In addition, a case had to have any two of the following obvious signs of severe obstruction: caput formation, Bandl's ring, sub-conjunctival hemorrhages and edematous vulva.

Controls were women admitted to the labour suit within the same 24-hour period in active labour without obstruction.

### Sample size and sampling

We used the formula described by Fleiss with a continuity correction to estimate the sample size[15]. The exposure factor was the proportion of pregnant women who attended < 4 ANC visits. We enrolled 270 cases and 270 controls based on the following assumptions: two-sided 95% confidence level, power of 95%, ratio 1:1 to detect an odds ratio of at least 2 for the risk of OL among pregnant women who attended < 4 ANC visits as the main exposure variable[16–18]. We further assumed that controls were like any other pregnant woman in Uganda who attended at least 4 ANC visits (60%) according to the Uganda demographic and health survey [9].

We consecutively enrolled all eligible incident cases between July 2018 and February 2019. We used simple random sampling to select one control from a list of admissions in active labour immediately after enrolling each case. Before recruitment, all respondents gave us written informed consent and pregnant adolescents below the legal age of 18 years were taken as emancipated minors[19]. We used unique study numbers issued at enrolment to identify each respondent.

### Inclusion criteria

Cases were women with OL carrying singleton, term pregnancies in cephalic presentation. Controls were women in active labour without obstruction carrying singleton, term pregnancies in cephalic presentation.

### Exclusion criteria

We excluded women with other obstetric emergencies such as antepartum haemorrhage, Preeclampsia and eclampsia (defined as elevated blood pressure of at least 140/90 mmHg, urine protein of at least 2+, any of the danger signs and fits), premature rupture of membranes and

intrauterine fetal death. We also excluded all women from outside the Hospital catchment area of 14 districts as either cases or controls.

## Study variables

The socio-demographic factors highlighted in the literature to predispose women to OL were the participant's age, marital status, occupation, level of education, the occupation and education level of the spouse as well as distance to the nearest health facility and the place of residence[10,12,17,20,21]. The obstetric factors were gravidity, number of ANC visits, having a delivery plan in place, a history of being referred from a lower health facility and use of herbal medications during labour[16,17]. Physical examination included the respondent's height and fetal birth weight. Our main exposure was the number of ANC visits attended as indicated on the ANC card, the other covariates were considered as confounders.

## Data collection

We used an interviewer-administered questionnaire running on an open data kit (ODK) platform. Trained research assistants (RA's) who are qualified midwives administered the questionnaire to all participants in the local dialect. We blinded all the RA's to the hypothesis of the study. Available records such as the antenatal cards, facility registers and case report files were reviewed by the RA's to crosscheck some of the verbal responses. The principal investigator (PI) would, on a daily basis access and review the data from the Google Aggregate server for completeness.

## Data management

The data was uploaded to a password protected server to which only the PI or his designee had access. Assisted by a statistician, the data was downloaded into an excel spreadsheet and exported to Stata version 14 for further cleaning and analysis.

## Data analysis

Baseline socio-demographic, physical and obstetric characteristics of the cases and controls were compared, to identify any differences. Normality of the continuous variables was tested for using the Shapiro-Wilk test. We summarised continuous variables using means and standard deviations. Whereas frequencies and percentages were used for the categorical variables. We used logistic regression (LR) to estimate Odds ratios, and 95% confidence intervals to examine the association between the number of ANC visits ($< 4$ Vs $\geq 4$) and the different socio-demographic, physical and obstetric covariates in bivariable and multivariable analysis. We included all factors that are known to confound the relationship between the frequency of ANC attendance and OL in the multivariable LR model, based on biological plausibility. In order to control for potential residual confounding due to factors that we had not previously hypothesized to be confounders, we also included those variables for which bivariable analysis returned a p-value equal to or less than 0.25. We reasoned that a cut-off of 0.25 would allow us to test the effect of any factors previously not known to have a confounding effect on the relationship between OL and the frequency of ANC attendance, without including those factors that were reasonably least likely [22]. Multicollinearity between explanatory variables was assessed using the variance inflation factor (VIFs), and they were all less than 1.5.

In the final adjusted multivariable model, we included all the statistically significant covariates (being a referral, a history of using herbal medicines, having a delivery plan, prime parity

and partner education level). Confounding was considered present, if the difference between the crude and adjusted OR was ≥ 10 percentage points[23,24].

### Ethical considerations

The Makerere University School of Medicine Research and Ethics Committee (#REC REF 2017–103) and the Uganda National Council for Science and Technology (HS217ES) approved the protocol. The Mbale Hospital Research and Ethics Committee (MRRH-REC IN-COM 00/2018) gave us administrative clearance. The hospital protocols were followed in management emergencies during the study.

## Results

### Characteristics of the study population

The respondents were generally young with a mean age of 24.5± 6 years, of average stature with a mean height of 160±8.2 cm and gave birth to babies of normal birth weight with a mean of 3.3± 0.4 Kg. Almost all (99%) respondents attended at least one ANC visit, mostly (96%) in public health facilities. Two-thirds (68%) of the respondents had no delivery plan in place. Majority of respondents resided in rural areas (84%) with no formal employment (89%) and almost one-half (44%) had used herbal medications during labour. The cases were younger (mean age 23.5±5.9 Vs 25.4±5.9), P-value <0.001 and shorter (159±8.2 Vs 161.4±7.4), P-value 0.011 than the controls (Table 1).

### Factors associated with OL

Maternal age, height, marital status, level of education, occupation and place of residence as well as the spouse's level of education and occupation were associated with OL. Obstetric factors such as prime parity, presence of an abnormal fetal heart rate, use of herbal medications in labour and history of being referred were associated with OL.

The odds of obstructed labour among referred women were 10 [crude odds ratio (COR) 9.69: 95% CI 5.79–16.21)] times the odds of obstructed labour among the women not referred. We found no association between OL and the number of ANC visits (COR 1.01, 95% CI: 0.73–1.41). The fetal birth weight among cases was 3.30±0.45 and 3.36±0.41 among controls and was not associated with OL. The odds of obstructed labour among married women was 0.6 times (COR 0.59 (0.35–0.97) the odds of obstructed labour among unmarried women (Table 2).

After adjusting for confounding (Table 3), these factors were independently associated with OL: having a partner with post primary education (AOR 0.57 95% CI: 0.33–0.98), being a referral from a lower health facility (AOR 6.80, 95% CI: 4.20–11.00), prime parity (AOR 2.15 95% CI: 1.26–3.66), use of herbal medicines in labour (AOR 2.43 95% CI: 1.50–3.64), having a delivery plan (AOR 0.56 95% CI: 0.35–0.90) and a fetal heart rate < 120 beats per minute (AOR 10.78 95% CI: 1.21–96.11).

## Discussion

We conducted a case control study using incidence density sampling to identify risk factors for OL in Mbale Hospital. We found that increased frequency of ANC attendance (< 4 Vs ≥ 4 ANC visits) was not protective against OL, contrary to our postulation. The risk factors for obstructed labour were prime parity, use of herbal medicines in labour, being a referral from a lower health facility, as well as having a low fetal heart rate (<120 beats per minute) at

**Table 1. Baseline characteristics of the participants.**

| Characteristic | Cases | Controls | Total |
|---|---|---|---|
| | n = 270 (100%) | n = 270 (100%) | N = 540 (100%) |
| **Age, years (SD)*** | 23.5 (5.9) | 25.4 (5.9) | 24.5 (6.0) |
| less than 20 | 80 (29.6) | 42 (15.6) | 122 (22.6) |
| 20 to 29 | 147 (54.4) | 165 (61.1) | 312 (57.8) |
| 30 and above | 43 (15.9) | 63 (23.3) | 106 (19.6) |
| **Mean height, cm (SD)*** | 159 (8.2) | 161 (7.4) | 160 (8.2) |
| less than 150 | 46 (17.0) | 22 (8.2) | 68 (12.6) |
| 150 and above | 224 (83.0) | 248 (91.9) | 472 (87.4) |
| **Mean weight, kg (SD)*** | 64.1 (10.1) | 65.3 (9.3) | 64.7 (9.8) |
| **Mean fetal birth weight, kg (SD)*** | 3.30 (0.5) | 3.36 (0.4) | 3.33 (0.4) |
| less than 2.5 | 6 (2.2) | 2 (0.7) | 8 (1.5) |
| 2.5 to 3.5 | 166 (61.4) | 134 (49.6) | 300 (55.6) |
| >3.5 | 98 (36.3) | 134 (49.6) | 232 (43.0) |
| **Mean fetal heart rate, bpm (SD)*** | 138 (13.7) | 136 (8.4) | 136 (15.2) |
| less than 120 | 15 (5.6) | 2 (0.7) | 17 (3.2) |
| 120 to 160 | 240 (88.9) | 264 (97.8) | 504 (93.3) |
| above 160 | 15 (5.6) | 4(1.5) | 19 (3.5) |
| **Marital status** | | | |
| Not Married | 46 (17) | 29 (10.7) | 75 (13.9) |
| Married | 224 (83.0) | 241 (89.3) | 465 (86.1) |
| **Education level of respondent** | | | |
| Primary | 139 (51.5) | 99 (36.7) | 238 (44.1) |
| Post primary | 131 (48.5) | 171 (63.3) | 302 (55.9) |
| **Occupation of respondent** | | | |
| House wife | 176 (65) | 164 (61) | 340(63) |
| Peasant farmer | 40 (15) | 21 (8) | 61 (11) |
| Salary earner | 31 (12) | 28 (10) | 59 (11) |
| Retail business | 23 (9) | 57 (21) | 80 (15) |
| **Place of Residence** | | | |
| Rural | 239 (89) | 212 (79) | 451 (84) |
| Urban | 31 (12) | 58 (22) | 89 (17) |
| **Distance to the nearest Health Unit** | | | |
| < 5 km | 205 (75.9) | 221 (81.8) | 426 (78.9) |
| ≥ 5 km | 65 (24.1) | 49 (18.2) | 114 (21.1) |
| **Education level of spouse** | | | |
| Primary | 126 (46.7) | 74 (27.4) | 200 (37) |
| Post primary | 144 (53.3) | 196 (72.6) | 340 (63) |
| **Occupation of spouse** | | | |
| Peasant farmer | 177 (65.6) | 143 (53) | 320(59.3) |
| Retail business | 44 (16.3) | 58 (21.5) | 102(18.9) |
| Income earner | 49 (18.2) | 69 (25.6) | 118(21.9) |
| **Gravidity** | | | |
| Prime gravida | 150 (55.6) | 79 (29.3) | 229 (42.4) |
| Gravida 2 to 4 | 85 (31.5) | 151 (55.9) | 236 (43.7) |
| Gravida 5+ | 35 (13.0) | 40 (14.8) | 75 (13.9) |
| **Number of ANC visits** | | | |
| < 4 ANC visits | 153 (56.7) | 152 (56.5) | 305 (56.6) |

*(Continued)*

**Table 1.** (Continued)

| Characteristic | Cases | Controls | Total |
|---|---|---|---|
| | n = 270 (100%) | n = 270 (100%) | N = 540 (100%) |
| ≥ 4 ANC visits | 117 (43.3) | 118 (43.5) | 234 (43.4) |
| **Health facility attended for ANC** | | | |
| Public health facility | 261 (96.7) | 258 (95.5) | 518 (96.1) |
| Private health facility | 9 (3.3) | 12 (4.5) | 21 (3.9) |
| **Have a delivery plan in place** | | | |
| Yes | 79 (29.3) | 93 (34.6) | 172 (31.9) |
| No | 191 (70.7) | 177 (65.4) | 368 (68.1) |
| **Used herbal medicines in labour** | | | |
| Yes | 161 (59.6) | 79 (29.3) | 240 (44.4) |
| No | 109 (40.4) | 171 (70.7) | 300 (55.6) |
| **Being a referral** | | | |
| Yes | 184 (68.2) | 45 (16.7) | 229 (42.4) |
| No | 86 (31.9) | 225 (83.3) | 311 (57.6) |
| **Source of referral** | | | |
| Public health facility | 174 (95) | 40 (89) | 214 (94) |
| Private health facility | 10 (5.4) | 5 (11.1) | 15 (6.5) |

Abbreviations: cm, centimetre; km, kilometre; kg, kilogram; bpm, beats per minute; ANC, antenatal care; SD, standard deviation.

* Values are given as mean ±SD or number (percentages) unless stated otherwise

enrolment. Having a delivery plan in place, an educated male partner and being married were protective of OL.

In this study, almost all the participants attended at least one ANC visit, which made the cases and controls similar on this particular characteristic. For instance, 43.3% of the cases and 43.5% of the controls attended four or more ANC visits. Despite this high level of utilisation of ANC services in mostly government public health facilities, being a referral from a lower health facility in active labour was independently associated with OL, implying that the quality of care at the lower health facilities may be substandard[25]. This could be attributed to the existing mismatch between the low staffing levels and high patient turnover that is common at public health facilities in Uganda[25,26]. Therefore, it is not surprising that OL was not associated with the frequency of ANC attendance in the current study. In a case control study among obstetric fistula patients in western Uganda, Barageine et al found no association between ANC attendance and obstetric fistula (a direct consequence of prolonged OL) [13]. On the contrary, several descriptive studies done in Nigeria and Ethiopia have found none utilisation of ANC services to be associated with OL[27,28]. It is likely that the effect of increased frequency ANC on OL is small and therefore another study with lager sample to study the effect of timing and number of individual ANC visits on OL, since it is known that frequent ANC visits especially in the last trimester prevents adverse obstetric outcomes[29]. The occurrence of OL and its squeal is influenced by delays due to a none functional referral system such as duration of labour before arrival to a health facility and taking > 4 hours to travel to a health facility for care [12,30,31]. The current study did not investigate the delays associated with OL, which was a limitation. Nonetheless, our finding of the odds of obstructed labour among referred women being seven times the odds of obstructed labour among non-referred has important implications because OL is an emergency that needs to be relieved in its early stages to prevent the associated morbidity and mortality. For public health, it may be a pointer to the

**Table 2. Factors associated with obstructed labour at bivariable analysis.**

| Characteristic | Crude OR | 95% CI | P- Value |
|---|---|---|---|
| **Mean age, years (SD)\*** | ⁻0.05 | ⁻0.08 - ⁻0.03 | 0.000 |
| less than 20 | 2.14 | 1.38–3.30 | 0.001 |
| 20 to 29 | 1 | | |
| 30 and above | 0.77 | 0.49–1.20 | 0.243 |
| **Mean height, cm (SD)\*** | ⁻0.03 | ⁻0.05 - ⁻0.01 | 0.011 |
| less than 150 | 1.78 | 0.83–3.82 | 0.137 |
| above 150 | 1 | | |
| **Mean weight, kg (SD)\*** | ⁻0.01 | ⁻0.03 - ⁻0.00 | 0.142 |
| **Mean fetal birth weight, kg (SD)\*** | ⁻0.34 | ⁻0.74 - ⁻0.05 | 0.088 |
| less than 2.5 | 2.42 | 0.48–12.19 | 0.284 |
| 2.5 to 3.5 | 1 | | |
| above 3.5 | 0.59 | 0.42–0.83 | 0.003 |
| **Mean fetal heart rate, bpm (SD)\*** | ⁻0.02 | ⁻0.00 - ⁻0.03 | 0.040 |
| less than 120 | 14.24 | 1.85–109.6 | 0.011 |
| 120 to 160 | 1 | | |
| above 160 | 4.11 | 1.34–12.54 | 0.013 |
| **Marital status** | | | |
| Married | 0.59 | 0.35–0.97 | 0.037 |
| Not Married | 1 | | |
| **Education level of respondent** | | | |
| Post primary | 0.55 | 0.39–0.67 | 0.001 |
| Primary | 1 | | |
| **Occupation of respondent** | | | |
| House wife | 1 | | |
| Peasant farmer | 1.77 | 1.00–3.14 | 0.048 |
| Salary earner | 1.03 | 0.59–1.79 | 0.912 |
| Retail business | 0.38 | 0.22–0.64 | 0.000 |
| **Place of Residence** | | | |
| Rural | 0.48 | 0.30–0.77 | 0.003 |
| Urban | 1 | | |
| **Distance to the nearest Health Unit** | | | |
| ≥ 5 km | 1.43 | 0.94–2.18 | 0.093 |
| <5 km | 1 | | |
| **Education level of spouse** | | | |
| Primary | 0.43 | 0.30–0.62 | 0.000 |
| Post primary | 1 | | |
| **Occupation of spouse** | | | |
| Peasant farmer | 1 | | |
| Retail business | 0.61 | 0.39–0.96 | 0.033 |
| Paid employee | 0.57 | 0.37–0.88 | 0.011 |
| **Gravidity** | | | |
| Prime gravida | 3.37 | 2.31–4.94 | 0.000 |
| Gravida 2 to 4 | 1 | | |
| Gravida 5+ | 1.55 | 0.92–2.63 | 0.100 |
| **Number of ANC visits** | | | |
| < 4 ANC visits | 1.01 | 0.73–1.41 | 0.933 |
| ≥ 4 ANC visits | 1 | | |

(*Continued*)

**Table 2.** (Continued)

| Characteristic | Crude OR | 95% CI | P- Value |
|---|---|---|---|
| **Health facility attended for ANC** | | | |
| Public health facility | 1.38 | 0.55–3.42 | 0.493 |
| Private health facility | 1 | | |
| **Have a delivery plan** | | | |
| Yes | 0.79 | 0.55–1.13 | 0.196 |
| No | 1 | | |
| **Used herbal medicines in labour** | | | |
| Yes | 3.65 | 2.45–5.42 | 0.000 |
| No | 1 | | |
| **Being a referral** | | | |
| Yes | 9.69 | 5.79–16.21 | 0.000 |
| No | 1 | | |
| **Source of referral** | | | |
| Public health facility | 1.00 | 0.20–4.96 | 1.000 |
| Private health facility | 1 | | |

Abbreviations: cm, centimetre; km, kilometre; kg, kilogram; bpm, beats per minute; ANC, antenatal care; SD, standard deviation; CI, confidence interval.

* logit coefficients.

lack of capacity to manage abnormal labour at district level hospitals and county level health centre IV's to offer emergency obstetric services closer to the community as it was envisioned in the governments' decentralisation plan[32]. Most of the patients were sent without clear documentation and specific diagnosis of obstructed labour. Sixty percent of the women with OL had used herbal medications in labour compared to 29% of the controls. Very often, when labour is not progressing well there is a high tendency to use local herbs in an attempt to quicken the process[33,34]. Referral to larger health facility is usually a last resort when everything else has failed[35]. So, it is not surprising that the odds of OL were two times higher among women with a positive history of having used herbal medications compared to those with a negative history.

The odds of obstructed labour were two times higher among the prime paras compared to the multiparous women in our study. Several studies have reported similar findings [2,12,36]. In our setting, many first time mothers are also young and it is possible that a link exists between prime parity and maternal age[2,11,12]. Although the current study was not powered to study this relationship, we know that young girls are prone to OL because they have an under developed pelvic cavity [2,13,37]. In addition, they have limited access to quality maternity services due to social and economic disadvantages and the fact that they usually conceive outside formal marriage. A prospective study involving only teenagers or prime paras would be necessary to resolve this contradiction.

Contrary to findings from similar low resource settings, the participants height, education level, occupation, distance to the nearest health facility with emergency obstetric care services and the occupation of the spouse were not identified as risk factors for OL [2,10–13,36]. Although, having an educated spouse (at least post primary level) and a delivery plan in place was protective of OL. Our findings are in agreement with the thinking that the known risk factors for OL have a poor predictive value, which makes primary prevention difficult [2,10,12,36]. This underscores the importance of having each child birth supervised by a skilled

**Table 3. Risk factors independently associated with obstructed labour.**

| Characteristic | Adjusted OR | 95% CI | P- Value |
|---|---|---|---|
| **Maternal age, years** | | | |
| less than 20 | 0.9 | 0.48–1.70 | 0.747 |
| 20 to 29 | 1 | | |
| 30 and above | 0.93 | 0.47–1.82 | 0.822 |
| **Maternal height, cm** | | | |
| above 150 | 1 | | |
| Less than 150 | 1.08 | 0.41–2.86 | 0.875 |
| **Fetal heart rate, bpm** | | | |
| less than 120 | 10.78 | 1.21–96.11 | 0.033 |
| 120 to 160 | 1 | | |
| above 160 | 2.37 | 0.62–9.01 | 0.205 |
| **Fetal birth weight, kg** | | | |
| less than 2.5 | 1.95 | 0.23–16.71 | 0.541 |
| 2.5 to 3.5 | 1 | | |
| above 3.5 | 0.95 | 0.606–1.49 | 0.818 |
| **Marital status** | | | |
| Not Married | 1 | | |
| Married | 0.92 | 0.47–1.78 | 0.796 |
| **Education level of respondent** | | | |
| Post Primary | 0.65 | 0.38–1.13 | 0.127 |
| Primary | 1 | | |
| **Occupation of respondent** | | | |
| House wife | 1 | | |
| Peasant farmer | 1.25 | 0.56–2.76 | 0.341 |
| Salaried | 1.50 | 0.68–3.31 | 0.318 |
| Retail business | 0.80 | 0.42–1.55 | 0.514 |
| **Place of Residence** | | | |
| Rural | 1.77 | 0.89–3.54 | 0.104 |
| Urban | 1 | | |
| **Distance to the nearest Health Unit** | | | |
| $\geq$ 5 km | 0.94 | 0.45–1.97 | 0.870 |
| <5 km | 1 | | |
| **Education level of spouse** | | | |
| Post Primary | 0.57 | 0.33–0.98 | 0.042 |
| Primary | 1 | | |
| **Occupation of spouse** | | | |
| Peasant farmer | 1 | | |
| Retail business | 1.44 | 0.73–2.84 | 0.291 |
| Paid employee | 1.19 | 0.68–2.10 | 0.537 |
| **Gravidity** | | | |
| Prime gravida | 2.15 | 1.26–3.66 | 0.005 |
| Gravida 2 to 4 | 1 | | |
| Gravida 5+ | 0.80 | 0.36–1.748 | 0.573 |
| **Number of ANC visits** | | | |
| < 4 ANC visits | 0.95 | 0.61–1.48 | 0.821 |
| $\geq$ 4 ANC visits | 1 | | |
| **Have a delivery plan in place** | | | |

(*Continued*)

**Table 3.** (Continued)

| Characteristic | Adjusted OR | 95% CI | P- Value |
|---|---|---|---|
| Yes | 0.56 | 0.35–0.90 | 0.017 |
| No | 1 | | |
| **Used herbal medicines in labour** | | | |
| Yes | 2.34 | 1.50–3.64 | 0.000 |
| No | 1 | | |
| **Being a referral** | | | |
| Yes | 6.80 | 4.20–11.00 | 0.000 |
| No | 1 | | |

Abbreviations: cm, centimetre; km, kilometre; kg, kilogram; bpm, beats per minute; ANC, antenatal care; SD, standard deviation; CI, confidence interval.

birth attendant. Although, the discrepancy might also be because we adopted an analytical approach to identify independent risk factors, while the earlier studies were mostly descriptive in nature to identify associated factors.

Fetal size was not a risk factor for OL. It is known that carrying a big baby (> 4kg) is a risk factor for OL because it increases the likelihood of cephalopelvic disproportion which is a common cause of OL [12,13]. In this study, the mean fetal birth weight was 3.33 kg and there was no significant difference between cases and controls on this characteristic. Ndibazza et al reported a mean fetal birth weight of 3.17 kg among 2,507 pregnant women recruited in a clinical trial in central Uganda [25], which is similar to our findings. In addition, most of the participants in this study were small with a mean body weight of 62 kg and no significant differences between cases and controls.

Post hoc power calculations suggest that our study may have been underpowered to detect a clinically important difference between the frequency of antenatal care visits (< 4 Vs ≥ 4 ANC visits) and OL even if the difference had been there (S1 File). For now, our results need to be interpreted with caution until they are validated by larger studies powered to detect small differences. However, conducting post hoc power calculations of this type may not be helpful as this can easily be seen from the confidence intervals that show an imprecise estimate and there is a huge body of statistical evidence that calculating a post hoc power is logically flawed (S3 File).

## Methodological considerations

In this study, we used incidence density sampling to identify controls. This strategy helped us to minimise selection bias but we could not assess the effect of time/ duration that has been highlighted as a risk factor in several other studies[12].

The RA's were well trained and blinded to the main hypothesis of the study to minimise information bias arising from paying more attention to the cases during the interviews. We triangulated the sources of information by supplementing the verbal responses with a review of the participant's case notes.

In this hospital-based study, most of the patients were referrals so the findings might not be a true representation of the picture in the Elgon sub-region. It would be interesting to compare the referred cases with controls selected from the same referring health facility, which was beyond the scope of this study. These results may be generalizable to other regional referral hospitals in Uganda because the health care delivery system is uniformly organised across the country.

## Conclusion

Prime parity, being a referral and history of using herbal medicines in labour were identified as risk factors for OL. On the other hand, having a delivery plan in place and an educated partner (at least post primary level) were found to be protective of OL. We found no association between the frequency of ANC attendance and the risk of OL.

## Supporting information

**S1 File. Post hoc power calculations.**
(DOCX)

**S2 File. Dataset RFOLMUK.** The data set is in Stata format 14 and includes all variables analysed for this manuscript.
(DTA)

**S3 File. Post hoc power calculations rebuttal.**
(DOCX)

## Acknowledgments

We thank the study participants for accepting to be part of the study and the research midwives for working tirelessly to accomplish this task namely Ms. Auma Prosscovia, Ms. Nandutu Sarah Waterah, Mrs. Atim Ketty Ojwar, Ms. Alibo Elizabeth, Ms. Sarah Talyewoya and Ms. Jessica Muduwa.

## Author Contributions

**Conceptualization:** Milton W. Musaba, Grace Ndeezi, Justus K. Barageine, Andrew Weeks, Victoria Nankabirwa, James K. Tumwine, Julius N. Wandabwa.

**Data curation:** Milton W. Musaba, Daniel Semakula.

**Formal analysis:** Milton W. Musaba, Justus K. Barageine, Felix Wamono, Daniel Semakula, Julius N. Wandabwa.

**Funding acquisition:** Grace Ndeezi, Andrew Weeks, Victoria Nankabirwa, James K. Tumwine.

**Investigation:** Milton W. Musaba, Grace Ndeezi, Justus K. Barageine, Andrew Weeks, Victoria Nankabirwa, James K. Tumwine, Julius N. Wandabwa.

**Methodology:** Milton W. Musaba, Justus K. Barageine, Victoria Nankabirwa, James K. Tumwine, Julius N. Wandabwa.

**Project administration:** Milton W. Musaba, Julius N. Wandabwa.

**Resources:** Milton W. Musaba, James K. Tumwine.

**Software:** Milton W. Musaba, Felix Wamono, Daniel Semakula.

**Supervision:** Grace Ndeezi, Justus K. Barageine, Andrew Weeks, Julius N. Wandabwa.

**Writing – original draft:** Milton W. Musaba, Grace Ndeezi, Andrew Weeks.

**Writing – review & editing:** Milton W. Musaba, Grace Ndeezi, Justus K. Barageine, Andrew Weeks, Victoria Nankabirwa, Felix Wamono, Daniel Semakula, James K. Tumwine, Julius N. Wandabwa.

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
