## [Decision Letter · Decision Letter 0]

2 Jan 2020

PONE-D-19-32285

Risk factors for obstructed labour in Eastern Uganda: a case control study.

PLOS ONE

Dear Dr Musaba,

Thank you for submitting your manuscript to PLOS ONE. After careful consideration, we feel that it has merit but does not fully meet PLOS ONE’s publication criteria as it currently stands. Therefore, we invite you to submit a revised version of the manuscript that addresses the points raised during the review process.

We would appreciate receiving your revised manuscript by Feb 16 2020 11:59PM. To enhance the reproducibility of your results, we recommend that if applicable you deposit your laboratory protocols in protocols.io, where a protocol can be assigned its own identifier (DOI) such that it can be cited independently in the future. For instructions see: http://journals.plos.org/plosone/s/submission-guidelines#loc-laboratory-protocols

We look forward to receiving your revised manuscript.

Kind regards,

Calistus Wilunda, DrPH

Academic Editor

PLOS ONE

Additional Editor Comments (if provided):

Please include post hoc power calculations to show the extent to which the study was powered to examine the link between the factors assessed and obstructed labour. This is in the light of the fact that this study was not well powered to assess the link between ANC attendance and the outcome. You can include this information as a supplementary file.

Journal Requirements:

Reviewers' comments:

Reviewer's Responses to Questions

**Comments to the Author**

1. Is the manuscript technically sound, and do the data support the conclusions?

Reviewer #1: Yes

Reviewer #2: Partly

2. Has the statistical analysis been performed appropriately and rigorously? 

Reviewer #1: Yes

Reviewer #2: I Don't Know

3. Have the authors made all data underlying the findings in their manuscript fully available?

Reviewer #1: No

Reviewer #2: Yes

4. Is the manuscript presented in an intelligible fashion and written in standard English?

Reviewer #1: No

Reviewer #2: Yes

5. Review Comments to the Author

Reviewer #1: In this study, the authors conducted a case-control study to explore the risk factors for obstructed labour (OL) in Eastern Uganda. This study is of great public health importance. Overall, the study was well planned and conducted, analysis was correctly performed, results carefully discussed and manuscript nicely written. The authors also addressed the study limitations. Several issues:

1) The full name of an abbreviation should be given at the first appearance. For example, Introduction section, 1st paragraph, Line 63, “PPH”.

2) Introduction, 2nd paragraph, Line 67, what do you mean by “improving nutrition of the girl child…?”

3) Results section, 1st paragraph, Line 199, “159±8.2 Vs 1161.4±7.4”, please check height is correctly presented.

4) The result for the main hypothesis is negative. There is the issue of multiple comparisons, so the conclusions should be drawn with caution.

5) The presentations of Table 2 and Table 3 should be improved. The reference category usually should be consistent, either at the top or at the bottom.

6) Typos and grammatical errors need to be checked and corrected.

Reviewer #2: While the study presents the results of original research, the statistics and other analyses still need to be described in more sufficient detail to know whether they were performed to a high technical standard. The conclusions need to be presented in a more robust fashion and provide more explanation of what the data support and results show.

Even though the article is presented is written in standard English, it could be improved with additional copy-editing for correct grammar usage. Revisions would be easier to track if authors had highlighted or underlined changes in the text.

6. PLOS authors have the option to publish the peer review history of their article (what does this mean?). If published, this will include your full peer review and any attached files.

Reviewer #1: No

Reviewer #2: No

---

## [Author Response · Author response to Decision Letter 0]

20 Jan 2020

20th/01/2020

To 

Dr. Calistus Wilunda

Academic Editor

PLOS ONE

Dear Dr. Calistus Wilunda

Re; Response to reviewers’ comments and resubmission of revised manuscript ID PONE-D-19-20983

Thank you for taking off time to review and provide feedback on this manuscript. Please receive the revised copy with specific responses and changes summarized in the table below. 

Reviewers comment Response to comment Line number

Academic editor

Please include post hoc power calculations to show the extent to which the study was powered to examine the link between the factors assessed and obstructed labour. This is in the light of the fact that this study was not well powered to assess the link between ANC attendance and the outcome. You can include this information as a supplementary file. Thank you so much for the comment. We agree with the reviewer that most likely we did not have power to detect a difference between the frequency of antenatal care visits (< 4 Vs ≥ 4 ANC visits) and obstructed labour even if the difference had been there. This can easily be seen from the confidence intervals that show an imprecise estimate. The post hoc power calculations have been done for all the variables studied; referral status, use of local herbs, having a delivery plan, number of ANC visits, prime parity, occupation of the spouse, education level of spouse, distance from the nearest health facility, place of residence, occupation of the spouse, participants level of education, marital status, fetal birth weight and height of respondent.

We however agree with a huge body of statistical evidence that calculating a post hoc power is logically flawed for the reasons highlighted in supplement 3

 Included as a separate file name post hoc power calculations (S1).

Lines 284 to 289, page 17.

Included as a separate file Post hoc power calculations rebuttal (S3)

Journal Requirements:

Comment 1;

Please ensure that your manuscript meets PLOS ONE's style requirements, including those for file naming. We have updated all these accordingly and feel that the manuscript meets PLOS ONE’s style requirements. NA

Comment 2.

 We note that you have indicated that data from this study are available upon request. PLOS only allows data to be available upon request if there are legal or ethical restrictions on sharing data publicly.

b) If there are no restrictions, please upload the minimal anonymized data set necessary to replicate your study findings as either Supporting Information files or to a stable, public repository and provide us with the relevant URLs, DOIs, or accession numbers. Thank you for the guidance, after consultations with the sponsor of the study and the Makerere University School of Medicine Research and Ethics Committee (SOMREC), we have uploaded the minimal anonymized data set necessary to replicate study findings as Supporting Information file (S2 file. Dataset RFOLMUK) NA 

Reviewer 1

Comment.

In this study, the authors conducted a case-control study to explore the risk factors for obstructed labour (OL) in Eastern Uganda. This study is of great public health importance. Overall, the study was well planned and conducted, analysis was correctly performed, results carefully discussed and manuscript nicely written. The authors also addressed the study limitations. Several issues: Thank you for the feedback. NA

Issue1; 

The full name of an abbreviation should be given at the first appearance. For example, Introduction section, 1st paragraph, Line 63, “PPH”. Thank you for the observation, this has been written out in full as postpartum hemorrhage. Line 56, page 4.

Issue 2;

Introduction, 2nd paragraph, Line 67, what do you mean by “improving nutrition of the girl child…?” This has been elaborated further in the manuscript Lines 61-63, page 4.

Issue 3;

Results section, 1st paragraph, Line 199, “159±8.2 Vs 1161.4±7.4”, please check height is correctly presented. Thank you for the observation, this has been corrected in the manuscript to “159±8.2 Vs 161.4±7.4”, Line 183, page 9.

Issue 4;

The result for the main hypothesis is negative. There is the issue of multiple comparisons, so the conclusions should be drawn with caution. This limitation has been highlighted in the discussion section Line 233 to 236, page 15

Issue 5;

The presentations of Table 2 and Table 3 should be improved. The reference category usually should be consistent, either at the top or at the bottom. All the tables have been reviewed and revised Table 1: Line 185 to186, page 9 to 11

Table2: Line 200 to201, page 11 to 12

Table 3: Line 209 to 210, page 13 to 14.

Issue 6;

Typos and grammatical errors need to be checked and corrected. These have been checked and corrected throughout the document NA

Reviewer 2;

Comment 1.

While the study presents the results of original research, the statistics and other analyses still need to be described in more sufficient detail to know whether they were performed to a high technical standard. The conclusions need to be presented in a more robust fashion and provide more explanation of what the data support and results show. The statistical analysis has been described in more detail and the data set is attached.

We have also added a post hoc power calculation.

Discussed the power limitations in more detail 

 Line 150 to 171, page 8 and 9.

Supplement file S1.

Lines 284 to 289, page 17.

Comment 2.

Even though the article is presented is written in standard English, it could be improved with additional copy-editing for correct grammar usage. This has been done throughout the document NA

Comment 3.

Revisions would be easier to track if authors had highlighted or underlined changes in the text Sorry for the inconvenience, we have included a revised copy of the manuscript with track changes. Throughout the document.

END

Sincerely,

Dr. Musaba W. Milton

Department of Obstetrics and Gynaecology

Mbale Regional Referral Hospital/ Busitema University

---

## [Decision Letter · Decision Letter 1]

27 Jan 2020

Risk factors for obstructed labour in Eastern Uganda: a case control study.

PONE-D-19-32285R1

Dear Dr. Musaba,

We are pleased to inform you that your manuscript has been judged scientifically suitable for publication and will be formally accepted for publication once it complies with all outstanding technical requirements.

With kind regards,

Calistus Wilunda, DrPH

Academic Editor

PLOS ONE

Additional Editor Comments (optional):

Reviewers' comments:

Reviewer's Responses to Questions

**Comments to the Author**

1. If the authors have adequately addressed your comments raised in a previous round of review and you feel that this manuscript is now acceptable for publication, you may indicate that here to bypass the “Comments to the Author” section, enter your conflict of interest statement in the “Confidential to Editor” section, and submit your "Accept" recommendation.

Reviewer #1: All comments have been addressed

2. Is the manuscript technically sound, and do the data support the conclusions?

Reviewer #1: Yes

3. Has the statistical analysis been performed appropriately and rigorously? 

Reviewer #1: Yes

4. Have the authors made all data underlying the findings in their manuscript fully available?

Reviewer #1: Yes

5. Is the manuscript presented in an intelligible fashion and written in standard English?

Reviewer #1: Yes

6. Review Comments to the Author

Reviewer #1: The authors have addressed all my concerns after the revision.

I suggest accept the manuscript for publication.

7. PLOS authors have the option to publish the peer review history of their article (what does this mean?). If published, this will include your full peer review and any attached files.

Reviewer #1: No

---

## [Editor Report · Acceptance letter]

30 Jan 2020

PONE-D-19-32285R1 

Risk factors for obstructed labour in Eastern Uganda: a case control study. 

Dear Dr. Musaba:

I am pleased to inform you that your manuscript has been deemed suitable for publication in PLOS ONE. Congratulations! Your manuscript is now with our production department. 

With kind regards,

on behalf of

Dr. Calistus Wilunda 

Academic Editor

PLOS ONE